# Valorization of Pomegranate Peel: Mechanisms and Clinical Applications in Irritable Bowel Syndrome Management

**DOI:** 10.3390/ijms26083530

**Published:** 2025-04-09

**Authors:** Yu Guo, Lu Wang, Jun-Qing Huang, Mu-Wen Lu, Song-Hong Yang

**Affiliations:** 1School of Pharmacy, Jiangxi University of Chinese Medicine, Nanchang 330004, China; guoy046@jnu.edu.cn; 2School of Traditional Chinese Medicine, Jinan University, Guangzhou 510632, China; jqhuang@jnu.edu.cn; 3Nutrition and Bromatology Group, Department of Analytical and Food Chemistry, Faculty of Sciences, Universidade de Vigo, 32004 Ourense, Spain; lu2024wang@outlook.com; 4Guangdong Provincial Key Laboratory of Nutraceuticals and Functional Foods, College of Food Science, South China Agricultural University, Guangzhou 510632, China

**Keywords:** pomegranate peel, network analysis, phenols, irritable bowel syndrome

## Abstract

Current disposal methods for pomegranate peel (PP) waste are inadequate, resulting in environmental pollution. Given PP’s therapeutic potential in alleviating irritable bowel syndrome (IBS), elucidating its bioactive mechanisms is critical to guide its development into dietary supplements and promote sustainable recycling. In this study, bioinformatics and network analysis were employed to identify active compounds, key targets, and signaling pathways associated with PP’s therapeutic effects. We identified 39 bioactive compounds (primarily polyphenols) and 106 key targets linked to IBS. Network analyses revealed that PP polyphenols mitigate oxidative stress and inflammation, modulate estrogen receptors to enhance gastrointestinal motility, and regulate ferroptosis. These findings underscore PP’s potential as a therapeutic agent for IBS and provide a framework for repurposing food-processing byproducts.

## 1. Introduction

Pomegranate (*Punica granatum* L.), a member of the Punicaceae family native to South Asia and the Middle East [1], is globally valued for its nutrient-rich arils with distinctive sweet-sour flavor [2]. However, its processing generates substantial waste, notably pomegranate peel (PP), which constitutes 40–50% of the fruit’s weight. Approximately 1.9 million tons of PP were discarded globally in 2017 [3], primarily due to its high phenolic content causing intense bitterness [4]. Conventional disposal methods (e.g., landfilling, incineration) exacerbate environmental pollution through toxic leachates [5] and greenhouse gas emissions [6], necessitating sustainable valorization strategies. Recent studies highlight PP’s potential as a bioactive reservoir for pharmaceutical and nutraceutical applications [7,8], aligning with circular economy principles to transform agro-waste into health-promoting resources [9].

Irritable bowel syndrome (IBS), a functional gastrointestinal disorder affecting 4.1% of the global population (Rome IV criteria) [10], manifests as abdominal pain and altered bowel habits. Subtype prevalence varies geographically, with diarrhea-predominant (IBS-D) and mixed (IBS-M) forms dominating in China [11,12]. Pathophysiological mechanisms involve visceral hypersensitivity, gut–brain axis dysregulation, and microbiota alterations [13], complicating therapeutic management. Current pharmacological interventions (e.g., antispasmodics, anticholinergics) often provide incomplete symptom relief [14], underscoring the need for novel multi-target therapies derived from natural sources.

The resurgence of traditional medicine has intensified interest in plant-based IBS treatments. Granati pericarpium (PP in Chinese pharmacopoeia) [15] has been empirically used since the Han Dynasty for diarrhea management [16], yet its molecular targets and mechanisms remain elusive. This study integrates network pharmacology and computational biology to elucidate PP’s bioactive compounds, therapeutic targets, and signaling pathways in IBS, thereby facilitating its development as evidence-based dietary supplements for gastrointestinal health management.

## 2. Results

### 2.1. Bioactive Compounds and Target Identification in Pomegranate Peel

A curated database of 46 pomegranate peel (PP) compounds was established through systematic literature mining (Appendix A). Pharmacokinetic screening identified 39 bioactive candidates (Appendix A) with predicted interactions across 259 molecular targets (Table 1; Appendix A).

### 2.2. Analysis of GEO Differentially Expressed Genes Data and Acquisition of Targets for IBS

Considering that the predominant syndrome type among Chinese IBS patients is diarrhea-predominant (IBS-D), two chips closely related to it were selected from the GEO database for differentially expressed gene (DEG) analysis. The GSE36701 chip contained rectal and colon biopsies from 21 healthy volunteers (40 samples) and 27 IBS-D patients (41 samples). The GSE14841 chip contained biopsies from four healthy volunteers and five IBS-D patients. It is noteworthy that both chips used the GPL570 platform to upload information, which made them comparable and thus improved the reliability of the results. Using the principle of |logFC| ≥ 0.5 and *p*-value ≤ 0.05 for DEG screening, 152 DEGs were found in GSE36701 and 202 DEGs were found in GSE14841, as shown in Figure 1. The details of DEGs are shown in Appendix A. After removing duplicates, we obtained 353 DEGs. Finally, by integrating these DEGs with those collected from the Gene Cards and OMIM databases, a total of 2298 IBS-related targets were obtained.

### 2.3. Network Construction

After intersecting the 259 potential active-compound-affected targets with the 2298 IBS-related targets, a total of 106 key targets were obtained (Appendix A). These targets may be key for treating IBS with PP. A compound–target (C-T) network was constructed to analyze how compounds in PP affect targets and improve IBS (Figure 2). The C-T network contains eight red nodes that are hub targets subject to potentially active compounds that can regulate them to improve IBS-D. Natural plants have multi-compound and multi-target action, so we conducted a preliminary analysis through the C-T network to reveal how the potentially active compounds in PP improve IBS.

Two parameters were used to determine the importance of these nodes: degree (the number of edges connected to the node) and betweenness centrality (BC). Nodes with higher degree values are closer to the network hubs, meaning they can help us focus on important compounds or targets. Quercetin (degree = 37) had the highest degree, followed by granatin A and kaempferol (degree = 31). Quercetin and kaempferol are widely found in nature, especially in the peel of various fruits and vegetables, and have anti-inflammatory and antioxidant activities [17]. Ellagic acid, a popular dietary supplement, shows vasodilation, anti-inflammatory, anticancer, and cholesterol-lowering activity [18]. These are all popular natural antioxidants, demonstrating the potential of PP for waste recycling. Granatin A can act on 30 targets, including TLR4, NOS2, STAT3, MTOR, NFKB1, and PTGS2 in this network, suggesting that it has great potential to improve IBS by inhibiting inflammation.

The C-T network indicates that the compounds in PP are mainly polyphenols, with little difference in their degree values, averaging 26.03. Therefore, BC was introduced to help characterize the importance of nodes in the C-T network. Generally, BC is positively correlated with degree, but in this C-T network, although the degree value of alkaloid compounds such as pelletierine, isopelletierine, and pseudopelletierine is slightly lower than other compounds, their BC value ranks higher. Pseudopelletierine had an effect on the PROC, MMP9, ACHE, TLR4, and other targets in the C-T network, suggesting that they also made a great contribution to PP inhibiting inflammation and improving IBS.

For targets, CTSD, NR3C2, TDP1, and F13A1 had higher degree and BC values. CTSD is a lysosomal aspartic protease that has been used as a marker of estrogen response [19]. It has many other functions, such as functions in cell proliferation, invasion, metastasis, and cancer angiogenesis [20]. The classic estrogen receptor ESR2 also exists in this network, which reminds us of the investigation report that the number of female IBS patients is higher than that of males, and estrogen receptors may play a certain role in the development of IBS [21]. As PP is one of the medicines used to treat diarrhea in China, the IBS-D-related targets in the C-T network are also worth paying attention to. For IBS-D, LGALS3, CISD1, MMP9, PROC, CHKA, SLC40A1, EPHX1, and MMP7 are core targets.

A protein–protein interaction (PPI) network consisting of 106 nodes and 483 edges was constructed to further investigate the possible relationships among key targets in order to better understand the therapeutic mechanism of PP in IBS, as shown in Figure 3A (the first 35 targets are truncated for display). In the PPI network, the target’s degree is proportional to its importance, and the outcomes can identify targets worth considering. Appendix A provides more information about the PPI network. In Figure 3B, HSP90AA1, STAT3, MMP9, MTOR, PTGS2, and TLR4 were identified as potential core targets.

### 2.4. Gene Ontology (GO) and Kyoto Encyclopedia of Genes and Genomes (KEGG) Enrichment Analysis

To characterize the biological roles of the 106 key targets, GO enrichment analysis (*p*_adjust_ < 0.01) was performed (Figure 4). The results implicated PP in modulating multiple biological processes, including response to lipopolysaccharide (GO: 0032496), regulation of wound healing (GO: 0061041), and cellular response to oxidative stress (Figure 4A). Cellular component (CC) terms were predominantly associated with membrane microdomains (e.g., membrane raft, GO: 0045121; membrane microdomain, GO: 0098857), suggesting PP’s influence on membrane-bound signaling complexes (Figure 4B). Molecular function (MF) annotations highlighted kinase-related activities, particularly protein tyrosine kinase activity (GO: 0004713) and transmembrane receptor protein kinase activity (GO: 0019199), consistent with PP’s putative role in regulating inflammatory and proliferative pathways (Figure 4C; Appendix A).

KEGG pathway analysis identified 67 enriched pathways (*p*_adjust_ < 0.01), with the top 20 including Toll-like receptor signaling (hsa04620), NF-κB signaling (hsa04064), and IL-17 signaling (hsa04657), reinforcing PP’s anti-inflammatory mechanism (Figure 5; Appendix A). Notably, pathways linked to biliary secretion, calcium signaling, and estrogen signaling (hsa04915) were also enriched, aligning with network predictions of estrogen receptor (ESR2) involvement in IBS pathophysiology. These findings underscore PP’s multi-pathway therapeutic potential, bridging inflammation, redox regulation, and hormone-mediated gut motility.

### 2.5. Molecular Docking

Studies have demonstrated that colon biopsies from IBS-D patients show elevated protein content of PTGS2, which increases PGE2 production and subsequently promotes visceral hypersensitivity—A condition commonly observed and critically indicative in IBS patients [22]. In addition, studies have demonstrated that NOS2 and PTGS2 are promising therapeutic targets for IBS management [23]. Concurrently, there is ample clinical and experimental evidence supporting the safety and beneficial effects of pedunculagin [24] and punicalin [25]. Interestingly, our analysis suggests possible interactions among these metabolites, a finding that warrants deeper investigation. Therefore, considering these compelling insights, we specifically selected these metabolites for molecular docking analyses in our review to explore additional potential mechanisms of action.

Here, molecular docking techniques were used to explore and simulate the possible binding models of pedunculagin and punicalin with PTGS2 and NOS2, respectively, and the best binding affinities energy for each interaction were generated, as shown in Figure 6 and Table 2. In this study, celecoxib (CAS No. 169590-42-5) was employed as an inhibitor of PTGS2, while 1400 W (CAS No. 180001-34-7) was used as an inhibitor of NOS2, aiming to evaluate the reliability of pedunculagin and punicalin in their interactions with proteins. The results indicate that each compound binds to its protein target through visible hydrogen bonding and strong electrostatic interactions. Furthermore, the binding pockets of these compounds are located in the same region as their positive controls, suggesting the reliability of the simulation results. Based on these findings, the strong interactions between these active compounds with their targets (PTGS2 and NOS2) are considered to be the basis of their effective biological activity. Therefore, from a computer simulation perspective, these results demonstrate the potential of these compounds to exert anti-inflammatory and antioxidant effects and to treat IBS by influencing their relevant targets, further validating the predictions in the C-T network.

## 3. Discussion

IBS, a chronic and multi-functional gastrointestinal disorder, poses substantial clinical challenges due to its heterogeneous pathophysiology and the absence of universally effective therapies [26]. Conventional pharmacological interventions often yield suboptimal outcomes, prompting increased exploration of natural bioactive compounds with favorable safety profiles. Among these, polyphenol-rich pomegranate processing by-product (PP) has emerged as a promising candidate for IBS management. Notably, PP exhibits elevated concentrations of flavonoids, proanthocyanidins, and phenolic acids compared to pomegranate pulp [27], with network pharmacology analyses identifying polyphenols as its primary bioactive constituents and alkaloids as secondary contributors.

Previous studies confirmed the beneficial anti-inflammatory and antioxidant effects of PP on IBS, speculating that ellagitannins might mediate these effects. Additionally, polysaccharides were reported to support and enhance these anti-inflammatory effects [7,8]. However, that study did not elucidate the specific metabolites responsible for the observed effects. Our work directly supports and extends this previous research by providing potential underlying mechanisms. Specifically, we highlight the metabolites (such as granatin A and granatin B) likely responsible for modulating targets and pathways like NF-κB and TLR4, thus offering constructive insights for future targeted investigations. In this review, we found the therapeutic potential of PP aligns with the multi-factorial nature of IBS pathogenesis. Oxidative stress, a central driver of IBS progression [28], is mitigated by PP-derived polyphenols such as luteolin, quercetin, and punicalagin. These compounds suppress oxidative stress markers (e.g., NOS2) and modulate apoptosis-related pathways (e.g., TLR4/p38MAPK) [29,30], corroborating their anti-inflammatory roles. Transcriptomic profiling of IBS-D patients revealed up-regulated LGALS3 and CISD1 in colonic tissue—targets implicated in ROS accumulation and iron dysregulation [31]. PP polyphenols may counteract these effects by downregulating LGALS3 (a proinflammatory galectin modulating cytokine release [32,33,34]) and CISD1, thereby disrupting the oxidative stress–inflammation cycle [35]. Mechanistically, PP inhibits inflammatory mediators (e.g., PTGS2, TLR4) and modulates critical pathways, including IL-17, NF-κB, and Toll-like receptor signaling, suggesting broad-spectrum anti-inflammatory activity.

Emerging evidence further implicates ferroptosis—An iron-dependent cell death pathway—In IBS pathophysiology [36]. We uniquely propose that PP may alleviate IBS through the inhibition of ferroptosis, a mechanism previously unexplored in PP research. Our integrated analysis of GEO datasets and protein–protein interaction (PPI) networks identified NRF2 (a master regulator of redox homeostasis, or so-called NFE2L2) as one of the central hubs in PP’s mechanism. Notably, NRF2 not only occupies a prominent position in the C-T network but also emerges as a paramount player among the top-ranked targets (TOP5, degree = 34), which governs ferroptosis via antioxidant gene regulation [37,38,39]. These findings position NRF2 activation as a key mechanism by which PP attenuates iron-induced oxidative damage in IBS. We have uniquely combined colonic transcriptomic data from the GEO database (comparing IBS-D patients versus healthy controls) with target prediction of active PP metabolites. This approach allowed us to identify IBS-specific targets (e.g., LGALS3, CISD1) and elucidate the role of core targets (e.g., HSP90AA1, STAT3) in IBS through PPI network analysis. Our methodology overcomes the limitations of traditional research that relies solely on in vitro experiments or single-omics data, providing a new paradigm for investigating the mechanisms of natural products.

Although the metabolites of pomegranate peel have been previously reported, our study fills critical gaps by revealing new mechanisms, particularly by systematically elucidating their multi-target synergistic mechanisms in the context of IBS for the first time. Through KEGG enrichment analysis, we first identified that PP polyphenols may regulate gastrointestinal motility via the estrogen signaling pathway (hsa04915). PP may also address IBS-associated gastrointestinal dysmotility. Estrogen receptor beta (ESR2), highly expressed in colonic epithelium, modulates intestinal motility and inflammation [40]. In vivo studies demonstrate that 17β-estradiol suppresses colonic contractility [41], while PP polyphenols (e.g., pseudopelletierine) may counteract this effect via ESR2 modulation. Additionally, pseudopelletierine’s predicted inhibition of acetylcholinesterase (AChE)—an enzyme limiting acetylcholine-mediated smooth muscle contraction [42,43,44]—suggests a dual mechanism for enhancing gastric motility in IBS. These findings provide potential explanations for the gender differences in IBS prevalence, a mechanism previously unexplored.

Despite PP’s therapeutic promise, safety considerations warrant attention. While acute toxicity studies report no adverse effects in mice at doses ≤2000 mg/kg [45,46], chronic exposure to high-dose punicalagin (≥12.5 mg/kg) induces hepatotoxicity in rats, marked by elevated liver enzymes and lipid peroxidation [47]. These findings underscore the need for dose optimization and rigorous preclinical evaluation of PP-based formulations.

## 4. Materials and Methods

### 4.1. Collection of the Candidate Compounds of PP

As much relevant information as possible was collected to establish a comprehensive local database of the compounds in PP. Keywords such as “Pomegranate Peel” and “Granati Pericarpium” were used to search for articles published in peer-reviewed journals in several search engines, including cnki, Web of Science, Google Scholar, PubMed, SciFinder, and ScienceDirect. Subsequently, the identified PP compounds were compiled, and a comprehensive database was established for this study.

### 4.2. Screening of Potential Active Compounds

All candidate compounds were comprehensively screened using the “Bioavailability Score” (BS) on the SwissADME database (www.swissadme.ch, accessed on 26 July 2023) to identify potential active compounds worthy of further research [48]. The BS is commonly regarded as an important and objective index for evaluating the internal quality of compounds and is directly proportional to the likelihood of their clinical application. The screening criterion was used to select compounds with a BS value greater than or equal to 0.10 as potential active compounds [49].

### 4.3. Prediction of the Relevant Targets of PP Potential Active Compounds

In this study, traditional target prediction calculation methods were used to link the chemical similarity of drug-like compounds with molecular targets and therapeutic approaches based on the principle of similar chemical structure [50]. Essentially, the compound–target interaction data were extracted from SuperTarget, ChEMBL, and BindingDB and were standardized and integrated to create datasets for target prediction. Drug–target prediction was performed by utilizing the two-dimensional similarity between the compounds and the ligands associated with their respective targets (target sets). To enable comparison across different target sets, the raw scores were defined as the sum of all Tanimoto coefficients greater than the threshold of 0.45 for each target set and were normalized by dividing the raw score by the number of ligands corresponding to the respective targets. Z-scores were then calculated using the following formula to evaluate the specificity of the prediction:$$Z_{A}=\frac{\left(\frac{{Rawscore}_{A}-\mu }{N_{A}})\exp(0.335ln(\right)}{\sigma }$$

Here, *A* denotes the target set; *N_A_* denotes the number of ligands in the target set; *μ* and *σ* represent random background noise in the database [51]. The Z-score value is positively correlated with the importance of the predicted outcome.

### 4.4. Acquisition of Targets for IBS

The Gene Cards (www.genecards.org, accessed on 26 July 2023) and OMIM (www.omim.org, accessed on 26 July 2023) databases were used to search for information on IBS. After removing duplicates, relevant targets of IBS were obtained. In addition, high-throughput data chips from colonoscopic biopsy samples of IBS-D patients and healthy volunteers obtained from the GEO database (www.ncbi.nlm.nih.gov/gds, accessed on 26 July 2023) were analyzed to identify differentially expressed genes (GES14841 and GSE36701).

### 4.5. Network Construction

The potential target predictions of active compounds were intersected with the disease-related targets, and the overlapping targets were considered key targets. The compound–target (C-T) network was constructed by linking the potential active compounds with the key targets. Additionally, the protein–protein interaction (PPI) network relationship between the key targets belonging to “Homo sapiens” was constructed using the STRING online database (https://cn.string-db.org/, accessed on 26 July 2023). Finally, visual processing and analysis of each network were carried out using Cytoscape 3.9.1 software.

### 4.6. Enrichment of Gene Ontology (GO) and Kyoto Encyclopedia of Genes and Genomes (KEGG) Pathways Analysis

Enrichment analysis and functional annotation clustering were performed using Bioconductor 3.17 (http://bioconductor.org/, released on 26 April 2023), which is compatible with R 4.3.0 (https://www.r-project.org/, released on 21 April 2023), for the 106 key targets. Statistically significant GO terms and related pathways were collected with a cutoff of *p*_adjust_ ≤ 0.01.

### 4.7. Molecular Docking

To investigate the potential interactions between active compounds and their targets at the molecular level and explore their binding modes for validating previous predictions, AutoDock 4.2.6 was used to perform molecular docking of the potential active compounds with their targets. The size of the grid box in AutoDock Vina was set to 40 × 40 × 40 with 0.05 nm between the grid points, and the energy range was kept at the default setting. The crystal structures of human PTGS2 (PDB ID: 5F19) and NOS2 (PDB ID: 4UX6) were obtained from the RCSB Protein Data Bank (PDB, www.rcsb.org, accessed on 26 July 2023). The program calculates based on the different binding energies of each ligand and yields nine possible conformations. After selecting the best model based on binding affinity energy and molecular contacts, the docked complexes were analyzed and visualized using PyMOL version 2.5.10 (www.pymol.org, accessed on 26 July 2023).

## 5. Conclusions

Pomegranate processing by-product (PP), an underutilized residue rich in polyphenolic compounds, demonstrates significant potential for managing irritable bowel syndrome (IBS). In this study, we employed a multi-faceted approach—including differential gene expression profiling, network pharmacology, and molecular docking simulations—to investigate the therapeutic mechanisms of PP-derived phenolic compounds. Our analyses revealed that these bioactive constituents exhibit dual antioxidant and anti-inflammatory properties, which may alleviate IBS-related pathophysiology. Furthermore, molecular docking results suggest that PP polyphenols could enhance gastrointestinal motility through estrogen receptor modulation while concurrently regulating iron homeostasis. These findings not only substantiate the clinical potential of PP as a functional food ingredient for IBS management but also highlight the broader value of reutilizing agro-industrial by-products in nutraceutical development.

## Figures and Tables

**Figure 1 ijms-26-03530-f001:**
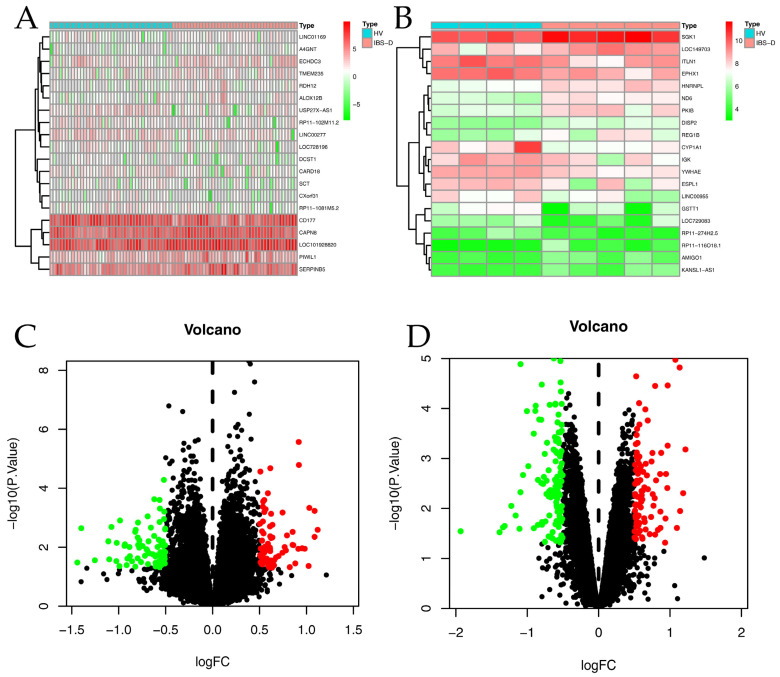
Analysis of GEO DEG data. (**A**) The heat map of the top 20 DEGs in the GSE36701 chip; (**B**) The heat map of the top 20 DEGs in the GSE14841 chip (green means low expression, white means medium expression, and red means high expression); (**C**) The volcano map of the GSE36701 chip; (**D**) The volcano map of the GSE14841 chip. Red means that the DEG was up-regulated in IBS-D compared to healthy volunteers (*p*-value ≤ 0.05 and logFC ≥ 0.5); Green means that the DEG was down-regulated in IBS-D compared to healthy volunteers (*p*-value ≤ 0.05 and logFC ≤ −0.5).

**Figure 2 ijms-26-03530-f002:**
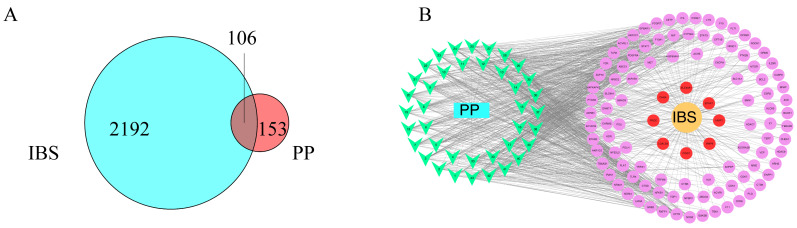
Compound–target network analysis. (**A**) The overlapping targets between PP and IBS. (**B**) The compound–target (C-T) network. The V-shaped nodes represent potential active compounds, the ellipse nodes represent the 106 overlapping key targets, and the red ellipse nodes represent the 8 hub targets of IBS-D.

**Figure 3 ijms-26-03530-f003:**
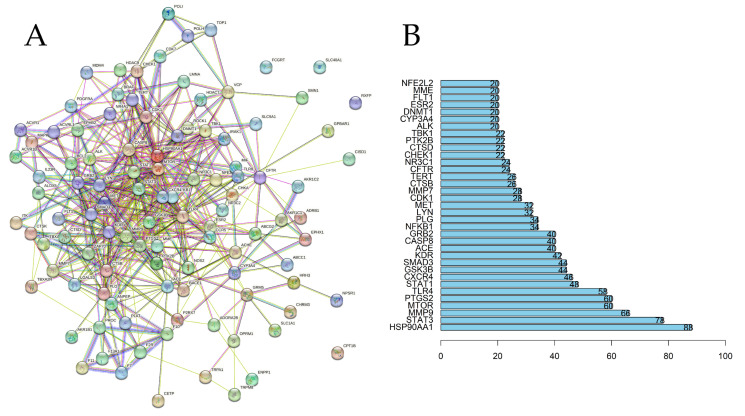
The protein–protein interaction (PPI) network analysis. (**A**) The PPI network relationship. (**B**) The number of targets that can interact with each other in the PPI network.

**Figure 4 ijms-26-03530-f004:**
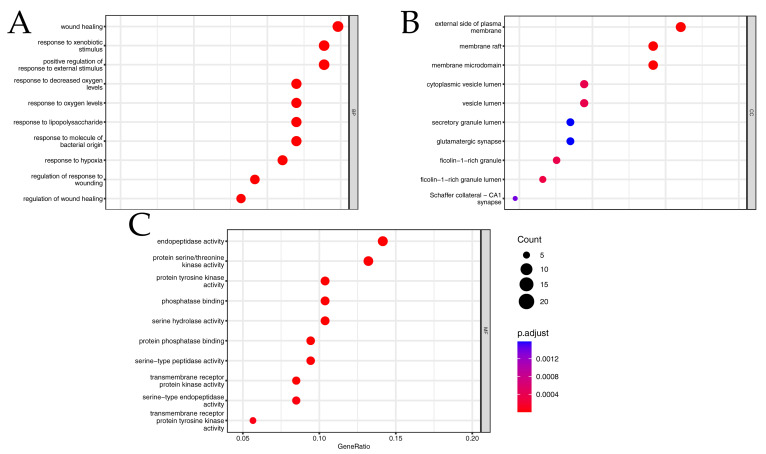
The results of GO enrichment analysis. (**A**) The biological process terms. (**B**) The cell component terms. (**C**) The molecular function terms.

**Figure 5 ijms-26-03530-f005:**
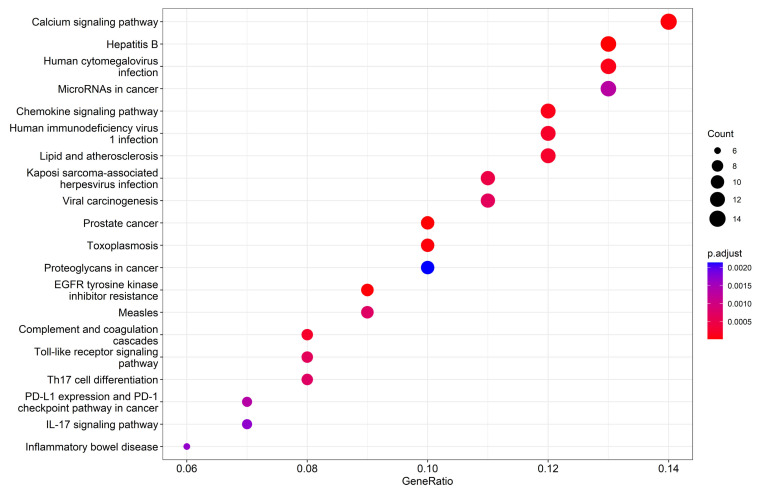
The results of KEGG pathway enrichment analysis.

**Figure 6 ijms-26-03530-f006:**
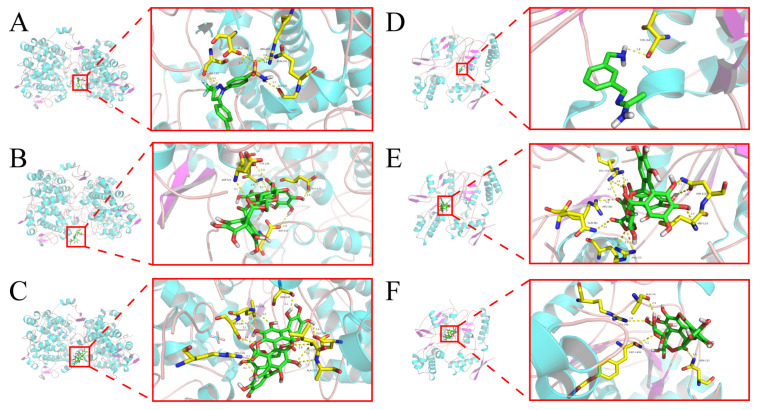
Molecular docking results of the possible binding models. (**A**) Celecoxib with PTGS2. (**B**) Pedunculagin with PTGS2. (**C**) Punicalin with PTGS2. (**D**) 1400 W with NOS2. (**E**) Pedunculagin with NOS2. (**F**) Punicalin with NOS2. The ball and stick model is used to represent molecules, while hydrogen bonds are depicted using dotted lines. Distances are measured in angstroms. The colors green, red, and blue are used to represent C, O, and N atoms, respectively.

**Table 1 ijms-26-03530-t001:** The potential active compounds of PP.

No.	Name	CAS	Class	No.	Name	CAS	Class
PP01	β-Glucogalin	13405-60-2	Polyphenols	PP21	Granatin B	77322-54-4	Polyphenols
PP02	Gallic acid	149-91-7	Polyphenols	PP22	Valoneic acid dilactone	60202-70-2	Polyphenols
PP03	Punicalin	65995-64-4	Polyphenols	PP23	Ellagic acid	476-66-4	Polyphenols
PP04	2-O-Galloylpunicalin	103488-45-5	Polyphenols	PP24	Kaempferol-3-O-β-D-glucopyranoside	480-10-4	Polyphenols
PP05	Pedunculagin	7045-42-3	Polyphenols	PP25	3-Glucosylquercetin	482-35-9	Polyphenols
PP06	Urolithin D	131086-98-1	Polyphenols	PP26	5-Hydroxymethylfurfural	67-47-0	Furfurals
PP07	Gallocatechin	970-73-0	Polyphenols	PP27	Pelargonidin	7690-51-9	Polyphenols
PP08	Punicalagin	65995-63-3	Polyphenols	PP28	3,3′-Di-O-methylellagic acid 4′-glucoside	51803-68-0	Polyphenols
PP09	(−)-Epigallocatechin	970-74-1	Polyphenols	PP29	Apigenin	520-36-5	Polyphenols
PP10	Sanguisorbic acid dilactone	82203-11-0	Polyphenols	PP30	5-hydroxymethylfuran-3-carboxylic acid	246178-75-6	Furfurals
PP11	Procyanidin B2	15514-06-4	Polyphenols	PP31	Cyanidin	13306-05-3	Polyphenols
PP12	Tellimagrandin I	79786-08-6	Polyphenols	PP32	Pelletierine	2858-66-4	Alkaloids
PP13	Urolithin A	1143-70-0	Polyphenols	PP33	Isopelletierine	4396-01-4	Alkaloids
PP14	Granatin A	161205-11-4	Polyphenols	PP34	Pseudopelletierine	552-70-5	Alkaloids
PP15	Methyl gallate	99-24-1	Polyphenols	PP35	Oleanic Acid	508-02-1	Triterpenoids
PP16	Catechin	154-23-4	Polyphenols	PP36	Kaempferol	520-18-3	Polyphenols
PP17	Casuarinin	79786-01-9	Polyphenols	PP37	β-Sitosterol	83-46-5	Triterpenoids
PP18	Corilagin	23094-69-1	Polyphenols	PP38	Luteolin	491-70-3	Polyphenols
PP19	Castalin	19086-75-0	Polyphenols	PP39	Quercetin	117-39-5	Polyphenols
PP20	(−)-Gallocatechin gallate	4233-96-9	Polyphenols				

**Table 2 ijms-26-03530-t002:** The best binding affinity for each interaction.

Compounds	Affinity Energy (kcal/mol)	
	PTGS2	NOS2
Pedunculagin	−8.4	−9.3
Punicalin	−9.2	−9.4
Celecoxib	−8.4	/
1400 W	/	−7.3

## Data Availability

The data presented in this study are available in the Appendix A of this article.

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
