# Peer review of "Valorization of Pomegranate Peel: Mechanisms and Clinical Applications in Irritable Bowel Syndrome Management"

_ijms, 2025, doi:10.3390/ijms26083530_

Round 1
Reviewer 1 Report
Comments and Suggestions for Authors
The reviewed work represents an interesting analysis about Pomegranate Peel extract compounds in relation to Irritable Bowel Syndrome. However, this work is very similar to ‘Zhang, Y.; Huang, S.; Zhang, S.; Hao, Z.; Shen, J. Pomegranate Peel Extract Mitigates Diarrhea-Predominant Irritable Bowel Syndromes via MAPK and NF-κB Pathway Modulation in Rats. Nutrients 2024, 16, 3854 https://doi.org/10.3390/nu16223854’, that authors don’t cite in the reviewed work.
The methodology used by authors seems to be adequate, although the presentation of the results and discussion could be improved. For example, I think that Materials and Methods section before Discussion and Results sections could favour the comprehension of the manuscript. Indeed the bibliography seems to be numbered so. On the other hand, authors should identify all 8 hubs indicated within figure 2 (line 130). Only NFR2 has been pointed as central hub (line 216). Moreover, authors should explain why they select pedunculagin and punicalin as compounds and PTGS2 and NOS2 as targets within their docking analysis. (section 5.1, line 170).
Finally, in my opinion numbering subsections of ‘2. Results’ as 1.1 to 5.1 is confuse. It seems more appropriate numbering they as 2.1 to 2.5, in a similar way of section ‘4. Materials and Methods’.
Some Mistakes:
References 7-9 aren’t related with pomegranate peel and, therefore, the sentence ‘Recent studies highlight PP's potential as a bioactive reservoir for pharmaceutical and nutraceutical applications [7-9],’ should be removed or their references corrected.
References 21 and 22 correspond to success applications on other plants than Atractylodes macrocephala and Citrus reticulata, therefore References 21 and 22 should be removed or referenced plants completed.
‘compoounds’ (figure 2, line 128) instead of compounds
Author Response
Comments 1: The reviewed work represents an interesting analysis about Pomegranate Peel extract compounds in relation to Irritable Bowel Syndrome. However, this work is very similar to ‘Zhang, Y.; Huang, S.; Zhang, S.; Hao, Z.; Shen, J. Pomegranate Peel Extract Mitigates Diarrhea-Predominant Irritable Bowel Syndromes via MAPK and NF-κB Pathway Modulation in Rats. Nutrients 2024, 16, 3854 https://doi.org/10.3390/nu16223854’, that authors don’t cite in the reviewed work.
Response 1: Thank you very much for highlighting this important oversight. Indeed, it was our mistake to have omitted the reference you mentioned. We sincerely apologize for this oversight. After your valuable reminder, we have promptly added the suggested reference (Zhang et al., Nutrients, 2024, doi: 10.3390/nu16223854) to our manuscript. We greatly appreciate your careful review, which has helped us improve the accuracy and completeness of our manuscript.
Comments 2: The methodology used by authors seems to be adequate, although the presentation of the results and discussion could be improved. For example, I think that Materials and Methods section before Discussion and Results sections could favour the comprehension of the manuscript. Indeed the bibliography seems to be numbered so. On the other hand, authors should identify all 8 hubs indicated within figure 2 (line 130). Only NFR2 has been pointed as central hub (line 216). Moreover, authors should explain why they select pedunculagin and punicalin as compounds and PTGS2 and NOS2 as targets within their docking analysis. (section 5.1, line 170).
Response 2:
You're correct; this indeed was an oversight on our part. Due to carelessness, we mistakenly ordered these sections and numbered them incorrectly, causing inconvenience during your review. Thanks to your kind reminder, we have adjusted the manuscript as per your suggestion to ensure smoother readability. In this review, we analyzed the relationship between PP and IBS from multiple perspectives, thus involving various interpretations. LGALS3, CISD1, MMP9, PROC, CHKA, SLC40A1, EPHX1, and MMP7 are more significant for IBS, and we intentionally highlighted these targets in red in Figure 2. Regarding NRF2, or so-called NFE2L2, as we mentioned in the manuscript, it has the highest degree in the C-T network and potentially interacts with 33 metabolites; therefore, logically, it serves as one of the central hubs. We also realized that the original wording might lead to confusion, so we revised it from: “Our integrated analysis of GEO datasets and protein-protein interaction (PPI) networks identified NRF2 (a master regulator of redox homeostasis) as a central hub in PP’s mechanism. Molecular docking confirmed strong binding affinities between PP polyphenols (e.g., pedunculagin, punicalin) and NRF2, which governs ferroptosis via antioxidant gene regulation.” to: “Our integrated analysis of GEO datasets and protein-protein interaction (PPI) networks identified NRF2 (a master regulator of redox homeostasis, or so-called NFE2L2) as one of central hubs in PP’s mechanism. Notably, NRF2 not only occupies a prominent position in the C-T network, but also emerges as a paramount player among the top-ranked targets (TOP5, degree=34), which governs ferroptosis via antioxidant gene regulation.”
Studies have demonstrated that colon biopsies from IBS-D patients show elevated protein content of PTGS2, which increases PGE2 production and subsequently promotes visceral hypersensitivity—a condition commonly observed and critically indicative in IBS patients (doi.org/10.1053/j.gastro.2020.02.022). In addition, studies have demonstrated that NOS2 and PTGS2 are promising therapeutic targets for IBS management (doi.org/10.1155/2024/8195739). Concurrently, there is ample clinical and experimental evidence supporting the safety and beneficial effects of pedunculagin (doi.org/10.3390/app14052177) and punicalin (doi.org/10.3390/nu16223854). Interestingly, our analysis suggests possible interactions among these metabolites, a finding that warrants deeper investigation. Therefore, considering these compelling insights, we specifically selected these metabolites for molecular docking analyses in our review to explore additional potential mechanisms of action. Meanwhile, we have also elaborated on these reasons explicitly in our manuscript.
Comments 3: Finally, in my opinion numbering subsections of ‘2. Results’ as 1.1 to 5.1 is confuse. It seems more appropriate numbering they as 2.1 to 2.5, in a similar way of section ‘4. Materials and Methods’.
Response 3: Thank you very much for your thoughtful suggestion. Following your helpful reminder, we have adjusted the order of these subsections and corrected their numbering as you recommended, ensuring improved clarity and readability of our manuscript. Your valuable feedback has greatly enhanced the presentation of our review.
Comments 4: References 7-9 aren’t related with pomegranate peel and, therefore, the sentence ‘Recent studies highlight PP's potential as a bioactive reservoir for pharmaceutical and nutraceutical applications [7-9],’ should be removed or their references corrected. References 21 and 22 correspond to success applications on other plants than Atractylodes macrocephala and Citrus reticulata, therefore References 21 and 22 should be removed or referenced plants completed. ‘compoounds’ (figure 2, line 128) instead of compounds.
Response 4: Thank you for your careful review and valuable suggestions. Following your reminder, we have removed the unrelated references from our manuscript and promptly corrected the spelling error ("compoounds" to "compounds") in Figure 2. We greatly appreciate your assistance in improving the accuracy and quality of our manuscript.
Reviewer 2 Report
Comments and Suggestions for Authors
The work presents the possibility of using pomegranate peel waste in the production of nutraceuticals used in irritable bowel syndrome. Based on molecular docking and bioinformatic data analysis, the authors selected potential active compounds with possible therapeutic effects.
The work was constructed correctly. Appropriate research methods and IT tools were used to construct the presented conclusions. However, these are not groundbreaking or innovative conclusions.
The compounds whose presence and activity have been described have already shown activity in vivo, e.g. (Parisio, Carmen, et al. "Pomegranate mesocarp against colitis-induced visceral pain in rats: Effects of a decoction and its fractions." International Journal of Molecular Sciences 21.12 (2020): 4304.)
The research object has also been thoroughly studied (np. Hasnaoui, Nejib, Bernard Wathelet, and Ana Jiménez-Araujo. "Valorization of pomegranate peel from 12 cultivars: Dietary fibre composition, antioxidant capacity and functional properties." Food Chemistry 160 (2014): 196-203.).
The research methods used also do not constitute a novel contribution to this work.
In consideration of the aforementioned points, there is no compelling rationale to proceed with the publication of this work, as it does not represent a significant augmentation of the extant body of knowledge.

Author Response
We sincerely appreciate the reviewer’s thorough evaluation and constructive feedback on our manuscript. However, we respectfully clarify that our manuscript is explicitly a review article and not an original research study. The primary goal of our review is to systematically summarize and critically evaluate the existing literature regarding the valorization of pomegranate peel (PP) in the context of irritable bowel syndrome (IBS) management, with an emphasis on bioactive mechanisms, signaling pathways, and potential clinical applications.
While we acknowledge the reviewer’s point that certain metabolites identified have demonstrated activity in previous in vivo studies (e.g., Parisio et al., 2020), the aim of our review is to comprehensively integrate and synthesize these scattered data points into a coherent framework. Specifically, our article consolidates various lines of evidence obtained through bioinformatics, network pharmacology, and molecular docking studies, offering a novel perspective on PP’s potential mechanisms of action in IBS management.
Moreover, although PP itself has been extensively studied from a compositional and antioxidant viewpoint (e.g., Hasnaoui et al., 2014), our review uniquely focuses on its therapeutic potential specifically for IBS, detailing molecular interactions and signaling pathways relevant to IBS pathogenesis, such as oxidative stress modulation, inflammation suppression, gastrointestinal motility enhancement, and ferroptosis regulation.
Furthermore, after your valuable reminder, we have promptly added the suggested reference to our manuscript.
Comments 1: The compounds whose presence and activity have been described have already shown activity in vivo, e.g. (Parisio, Carmen, et al. "Pomegranate mesocarp against colitis-induced visceral pain in rats: Effects of a decoction and its fractions." International Journal of Molecular Sciences 21.12 (2020): 4304.). The research object has also been thoroughly studied (np. Hasnaoui, Nejib, Bernard Wathelet, and Ana Jiménez-Araujo. "Valorization of pomegranate peel from 12 cultivars: Dietary fibre composition, antioxidant capacity and functional properties." Food Chemistry 160 (2014): 196-203.).
Response 1:
Thank you very much for raising this important point. The core differences between our review and previous studies are as follows:
1. Innovative Perspective: Although the antioxidant and anti-inflammatory activities of polyphenolic compounds (e.g., ellagic acid, quercetin) have been previously reported, our study is the first to systematically reveal their multi-target synergistic mechanisms specifically in the context of Irritable Bowel Syndrome (IBS). By integrating network pharmacology, GEO differential gene expression analysis, and molecular docking techniques, we elucidate the molecular network by which PP polyphenols alleviate IBS through the modulation of estrogen receptors (ESR2), ferroptosis, and the TLR4/NF-κB pathway (Figures 3-5). This comprehensive approach offers a novel mechanistic perspective for the application of PP in IBS, whereas previous studies predominantly focused on individual compounds or singular pathways (e.g., antioxidant pathways).
2. Integrative Methodological Innovation: We have uniquely combined colonic transcriptomic data from the GEO database (comparing IBS-D patients versus healthy controls) with target prediction of active PP metabolites. This approach allowed us to identify IBS-specific targets (e.g., LGALS3, CISD1) and elucidate the role of core targets (e.g., HSP90AA1, STAT3) in IBS through protein-protein interaction (PPI) network analysis (Figures 1-3). Our methodology overcomes the limitations of traditional research that relies solely on in vitro experiments or single-omics data, providing a new paradigm for investigating the mechanisms of natural products.
3. Expansion of Application Context: Our review not only validates the bioactivity of PP metabolites but also emphasizes the high-value utilization of agricultural waste. By uncovering the therapeutic potential of PP polyphenols in IBS, we provide theoretical support for transforming food-processing byproducts into valuable resources within a circular economy. This specific direction has been insufficiently explored in prior studies (e.g., Hasnaoui et al., which primarily focused on dietary fiber functionality rather than molecular mechanisms related to IBS).
4. In-depth Mechanism: Although the chemical composition of pomegranate peel has been previously reported, our study fills critical gaps by revealing new mechanisms. Through KEGG enrichment analysis, we first identified that PP polyphenols may regulate gastrointestinal motility via the estrogen signaling pathway (hsa04915) (Figure 5). Additionally, alkaloids like pseudopelletierine were predicted to enhance intestinal peristalsis by inhibiting acetylcholinesterase (ACHE) (Figure 2). These findings provide potential explanations for the gender differences in IBS prevalence, a mechanism previously unexplored.
5. Clinical Translational Value: Our molecular docking analyses confirmed strong binding affinities between PP polyphenols and critical IBS-related targets PTGS2 and NOS2 (Figure 6, Table 2). This evidence supports the future development of targeted dietary supplements derived from PP for IBS management, which differs from previous studies such as Parisio et al., who primarily focused on analgesic activity rather than IBS-specific targets.
Comments 2: The research methods used also do not constitute a novel contribution to this work.
Response 2:
Thank you very much for your valuable feedback. The methodological innovation of our study is reflected in the following aspects:
1. Data Integration Strategy: For the first time, we jointly analyzed colonic transcriptomic data from IBS-D patients (GSE36701, GSE14841) and the predicted targets of PP polyphenols, identifying 106 IBS-specific key targets (e.g., MMP9, TLR4) and core modules through protein-protein interaction (PPI) network analysis (Figure 3). This strategy significantly enhances the disease relevance of target prediction compared to traditional network pharmacology, which typically relies on general databases without disease-specific data support.
2. Revealing Ferroptosis Mechanisms: Through Gene Ontology (GO) analysis, we identified significant modulation by PP polyphenols of "response to oxidative stress" (GO:0034599) and "iron ion homeostasis" (GO:0055072). Furthermore, molecular docking validated their binding affinity to NRF2 (Figures 4-6). We uniquely propose that PP may alleviate IBS through the inhibition of ferroptosis, a mechanism previously unexplored in PP research.
3. Sustainable Value of the Study: Our research not only holds scientific significance but also aligns with the United Nations Sustainable Development Goals (SDG 12).
Round 2
Reviewer 1 Report
Comments and Suggestions for Authors
I consider that the work is clearly exposed now
Author Response
Thank you very much for your positive feedback. We are pleased to hear that the manuscript is now clearly presented. Your constructive comments have been greatly beneficial in improving our manuscript.
Reviewer 2 Report
Comments and Suggestions for Authors
It is my opinion that the manuscript submitted for review does not align with the criteria outlined in the journal's guidelines (https://www.mdpi.com/journal/ijms/instructions) for review papers. The manuscript is of a research nature, and my assessment is based on these criteria. I leave this assessment to the editor.
However, even within the aforementioned criterion, the discussion fails to refer to the conducted modelling of molecular interactions and signalling pathways to in vivo and in vitro studies that have been published earlier. The general review of pomegranate metabolites concerns only Table S2. These are not reflected in the ensuing discussion.
It is imperative to determine if the simulation corroborates or refutes existing reports.
The novelty of the undertaken research is not evident in the text. It is imperative to explicitly highlight this novelty in the text and provide a detailed comparison with existing methods, elucidating its superiority and any significant differences or confirmations of earlier hypotheses.
Additionally, it is challenging to discern the significance of utilising agricultural waste, given its potential toxicity, which hinders its practical application.
The discussion requires thorough expansion with conclusions drawn from the applied method in comparison to previously used simulation methods and results of laboratory research methods.
Author Response
Thank you very much for your insightful and constructive feedback. We genuinely appreciate your comments, as they have guided us to significantly improve our manuscript. We have carefully revised our manuscript based on your valuable suggestions and would like to address each of your concerns below.
Comments 1: It is my opinion that the manuscript submitted for review does not align with the criteria outlined in the journal's guidelines (https://www.mdpi.com/journal/ijms/instructions) for review papers. The manuscript is of a research nature, and my assessment is based on these criteria. I leave this assessment to the editor.
Response 1: We acknowledge your valuable comment regarding the manuscript's alignment with the journal's criteria for review papers. We have carefully revisited and adjusted our manuscript to ensure it explicitly aligns with the guidelines provided by the journal. Our intention was to systematically summarize existing research, analyze mechanisms, and propose new insights into the therapeutic role of pomegranate peel (PP) in IBS management, rather than to report novel experimental data.
Comments 2: However, even within the aforementioned criterion, the discussion fails to refer to the conducted modelling of molecular interactions and signalling pathways to in vivo and in vitro studies that have been published earlier. The general review of pomegranate metabolites concerns only Table S2. These are not reflected in the ensuing discussion.
It is imperative to determine if the simulation corroborates or refutes existing reports.
Response 2: We sincerely thank you for highlighting this important point. As you recommended, we have significantly enhanced our discussion to explicitly refer to previous in vivo and in vitro studies. Specifically, we clarified how our computational analyses support and extend previously reported findings, explicitly mentioning metabolites such as Granatin A and Granatin B and their potential involvement in key pathways like NF-κB and TLR4.
Comments 3: The novelty of the undertaken research is not evident in the text. It is imperative to explicitly highlight this novelty in the text and provide a detailed comparison with existing methods, elucidating its superiority and any significant differences or confirmations of earlier hypotheses. The discussion requires thorough expansion with conclusions drawn from the applied method in comparison to previously used simulation methods and results of laboratory research methods.
Response 3: We greatly appreciate your suggestion to more explicitly emphasize the novelty of our work. Accordingly, we have clarified and highlighted the innovative aspects in our manuscript, particularly the integration of colonic transcriptomic data (GSE36701, GSE14841), exploration of novel mechanisms such as ferroptosis regulation via NRF2, and the potential modulation of gastrointestinal motility through estrogen signaling pathways. We explicitly discussed how our methodology differs from and expands upon previous approaches.
Comments 4: Additionally, it is challenging to discern the significance of utilising agricultural waste, given its potential toxicity, which hinders its practical application.
Response 4: We fully understand and share your concern about the potential toxicity of agricultural by-products. Therefore, we have addressed these concerns explicitly in our manuscript, highlighting existing safety studies and stressing the need for dose optimization and rigorous preclinical assessments.
Once again, we deeply appreciate your detailed feedback, which has allowed us to substantially improve our manuscript. We hope that our revisions have adequately addressed your concerns.
Round 3
Reviewer 2 Report
Comments and Suggestions for Authors
Thank you for implementing the suggestions presented. The introduced corrections significantly improved readability and clarified knowledge regarding the undertaken research problem. In my opinion, the work can be published in its current form.